# The Impact of COVID-19 on Physician Burnout Globally: A Review

**DOI:** 10.3390/healthcare8040421

**Published:** 2020-10-22

**Authors:** Shabbir Amanullah, Rashmi Ramesh Shankar

**Affiliations:** 1Department of Psychiatry, University of Toronto, Toronto, ON M5S, Canada; 2Psychiatry Woodstock General Hospital, Woodstock, ON N4V 0A4, Canada; 3Norwich Medical School, University of East Anglia, Norwich NR4 7TJ, UK; rashmi22.shankar@gmail.com

**Keywords:** physician, health, burnout, pandemic

## Abstract

*Background*: The current pandemic, COVID-19, has added to the already high levels of stress that medical professionals face globally. While most health professionals have had to shoulder the burden, physicians are not often recognized as being vulnerable and hence little attention is paid to morbidity and mortality within this group. *Objective*: To analyse and summarise the current knowledge on factors/potential factors contributing to burnout amongst healthcare professionals amidst the pandemic. This review also makes a few recommendations on how best to prepare intervention programmes for physicians. *Methods*: In August 2020, a systematic review was performed using the database Medline and Embase (OVID) to search for relevant papers on the impact of COVID-19 on physician burnout–the database was searched for terms such as “COVID-19 OR pandemic” AND “burnout” AND “healthcare professional OR physician”. A manual search was done for other relevant studies included in this review. *Results*: Five primary studies met the inclusion criteria. A further nine studies were included which evaluated the impact of occupational factors (*n* = 2), gender differences (*n* = 4) and increased workload/sleep deprivation (*n* = 3) on burnout prior to the pandemic. Additionally, five reviews were analysed to support our recommendations. Results from the studies generally showed that the introduction of COVID-19 has heightened existing challenges that physicians face such as increasing workload, which is directly correlated with increased burnout. However, exposure to COVID-19 does not necessarily correlate with increased burnout and is an area for more research. *Conclusions*: There is some evidence showing that techniques such as mindfulness may help relieve burnout. However, given the small number of studies focusing on physician burnout amidst a pandemic, conclusions should be taken with caution. More studies are needed to support these findings.

## 1. Introduction

The medical profession is deemed to be a demanding and stressful profession with serious consequences if there is flawed decision making as it impacts patient care. The concept of burnout, which has been defined as a “psychological syndrome characterised by emotional exhaustion, depersonalisation and a sense of reduced accomplishment in day to day work” [1], is being increasingly recognised as a factor not only affecting physician health but also the patients in their care. Numerous previous studies have reported the huge prevalence of burnout seen amongst physicians [2,3,4,5], with one such study conducted by Shanafelt et al. reporting that 45.8% of a sample of 7288 US physicians had experienced burnout [6]. The consequences of burnout are potentially very serious for physicians as well as those with whom they interact. Burnout has been proven to cause a deterioration in the quality of care or services provided by the staff [1]. Furthermore, burnout appears to be correlated with increased use of alcohol and drugs, physical exhaustion and marital and family problems [1]. Hence, results reported by Shanafelt et al. are of huge concern, and with this current pandemic of COVID-19, it has only made it even more worrying and difficult for medical professionals.

To our knowledge, there have been very few papers that have explored the impact of COVID-19 on physician burnout, however, the method of assessing burnout has varied. The Maslach Burnout Inventory (MBI) is most widely used in research to measure burnout and has been regarded as the “measure of choice” for assessment of burnout [1]. It has been designed to assess the three components of burnout: emotional exhaustion, depersonalisation and reduced personal accomplishment—a score is then received for each component, which can be classified as low, average or high burnout status [1]. One such study by Wu et al. found that a significant proportion of physicians are experiencing more burnout after the introduction of COVID-19 compared to pre-COVID-19 using the Maslach Burnout Inventory (MBI) [7]. Whereas a cross-sectional survey by Ruiz-Fernández et al. used the “Professional Quality of Life Questionnaire” (ProQoL) to assess burnout and found that burnout levels amongst Spanish healthcare professionals have remained similar to those studies prior to the pandemic despite the health crisis situation [8]. There could be a number of different causes for the difference in results, for instance, the month in which the data was collected, the situation of the pandemic at the time of collection of data in each country, the method of assessing burnout etc. Hence, to draw conclusions on burnout amidst the pandemic, papers analysed in this systematic review were grouped together based on the assessment method of burnout.

Due to the uncertainty of the length of the current pandemic, one can only speculate the lasting impact to be considerable. Hence, it is important to address the issues that are leading to increased burnout during this pandemic in order to reduce the long-term negative consequences. To date, there are very few evidence-based interventions in literature that focus on physician burnout during a pandemic. However, a few studies have made recommendations that may help prevent burnout and mitigate the consequences of occupational stress during COVID-19 [9,10]. More studies are required to corroborate existing findings.

To date, there have been no reviews that have examined this area of literature during the COVID-19 pandemic, hence, this would be the first review to summarise the existing findings on the impact of the pandemic on physician burnout and provide a detailed analysis of the various identified and potential factors contributing to physician burnout. This paper also aims to make certain recommendations that may help relieve the effects of burnout during the course of the COVID-19 pandemic.

## 2. Methods

In order to review relevant publications for our review, the search was broadly split into three categories:(a)COVID-19 papers on physician burnout,(b)pre-COVID-19 papers on physician burnout,(c)support/recommendations for physician burnout before and after COVID-19.

### 2.1. Search Strategy

#### 2.1.1. COVID-19 Papers on Physician Burnout

This part of the systematic review was performed following PRISMA guidelines (Preferred Reporting Items for Systematic Review and Meta-Analyses). In order to search for papers on physician burnout relevant to COVID-19, we did a thorough literature search using the database Medline and Embase (OVID). Databases that were used for the search include Embase and Medline. The search algorithm that was applied can be seen in Table 1. In August 2020, studies that were eligible were selected through a multi-step approach (i.e., full text available, abstract, title reading). Initially, no articles were restricted by the process of filtering, to allow search of all available articles. 

#### 2.1.2. Pre-COVID-19 Papers on Physician Burnout 

A manual search was done for the relevant articles included in this final review based on the subheadings included in this review: “Occupational Factors, Gender Difference, Increased Workload/Sleep Deprivation.” This search was also conducted in August 2020.

#### 2.1.3. Support/Recommendations for Physician Burnout Before and After COVID-19

A manual search was done for the reviewed articles included in this final review based on the terms “Support”, “Recommendations”, “Physician Burnout” and “COVID-19”. This search was also conducted in August 2020.

### 2.2. Study Selection & Eligibility Criteria

#### 2.2.1. COVID-19 Papers on Physician Burnout

The first selection was performed by filtering duplicates. Next, titles and abstract were screened. Subsequently, all potentially relevant articles were then reviewed individually for eligibility. Studies were included if participants included were healthcare professionals who were involved in the care of patients, whilst those whose participants were not in clinical care were excluded. Secondly, for our first search we only focused on studies during COVID-19. Thirdly, experimental groups that did not include a control group were still included in our study. Fourth, in our search, studies using the MBI were separated from studies that used other assessors of burnout; this is because MBI is regarded as the gold standard for measuring burnout and results could be easily compared; however, there were very limited studies hence we didn’t exclude studies that didn’t contain MBI. Fifth, cross-sectional studies were included. Finally, only studies published in English were included. Figure 1 shows the flow chart of the selection of articles. Studies mentioned can be seen in Table 2.

#### 2.2.2. Pre-COVID-19 Papers on Physician Burnout 

As authors, we set general inclusion and exclusion criteria: (1) the results of the studies had to be published in English, (2) we excluded all articles where the abstract lacked information or the full text was not available and (3) we excluded articles that were not peer reviewed. As we searched papers prior to COVID-19 manually, we as authors chose what we thought were the most relevant papers related to the subject. In addition to the general selection criteria, for papers prior to COVID-19 on physician burnout, we decided to only consider publications from the year 2000 onward and update results to June 2020. In total, we chose nine studies for this section, which can be seen in Table 3.

#### 2.2.3. Support/Recommendations for Physician Burnout Before and After COVID-19

In addition to the general selection criteria, for papers mentioning support and recommendations for burnout prior to COVID-19, we decided to only consider publications from the year 2000 onward and update results to June 2020 to make this consistent with papers assessed in category (b). For papers offering support and recommendations for burnout during this current pandemic, we only accepted 2020 publications. In total, we chose five reviews for this section, which can be seen in Table 4.

### 2.3. Data Extraction and Quality Assessment

The same selection strategy was used for data extraction. As authors, we gathered and analysed the following information: study design (only cross-sectional studies and reviews for recommendations) were included; year of publication (2000–2020); outcomes. After analysing the full text, the target population included any healthcare professional that was involved in the clinical care of patients.

### 2.4. Quality Assessment

In order to assess the risk of bias, we used the Cochrane Manual for Systematic Intervention Review [22] as a guide; any discrepancies were discussed and agreed between the authors.

In the end, only five studies that explored the impact of COVID-19 on physician burnout were included, with a further nine studies exploring pre-identified factors contributing to burnout prior to COVID-19, and five reviews that offered recommendations for physician burnout were also analysed.

## 3. Physician Burnout during the COVID-19 Pandemic

### 3.1. Using the MBI to Measure Physician Burnout during COVID-19

For the first part of the evaluation of physician burnout in this review, the Maslach Burnout Inventory (MBI) was used. The MBI questionnaire contains 22 items that were designed to evaluate three particular components of burnout: emotional exhaustion, depersonalisation and personal accomplishment [1]. To our knowledge, there have only been three studies so far that have explored physician burnout during this pandemic using the MBI.

Wu et al. first explored the prevalence of burnout amongst medical staff in China, when China was the epicenter of the virus [7]. This study surveyed 220 physicians, with an equal split between males and females. All participants were given the Maslach Burnout Inventory–medical personnel (MBI-HSS(MP)) to complete. The results recorded an 86% response rate [7]. When individually asked about their attitudes towards COVID-19, the results demonstrated that only 23% of physicians had felt more burnout compared with before the COVID-19 crisis, 15% of respondents neither disagreed nor agreed to feeling ‘more burnout’ and 62% disagreed to feeling more burnout [7]. Results from MBI suggested that almost 25% of the sample felt increased emotional exhaustion and depersonalisation, with almost half of the participants reporting decreased personal accomplishment [7]. In comparison, another study was conducted by Guisti et al. who surveyed healthcare professionals working in a health institution in Northern Italy [13]. This study included a sample of 330 healthcare professionals who took part in the online survey that assessed burnout using the Maslach burnout Inventory-Human Service Survey (MBI-HSS). Results showed that more than two-thirds of participants had reported moderate to severe levels of emotional exhaustion and reduced personal accomplishment, and more than a quarter of the sample reported moderate to severe levels of depersonalisation. This level of physician burnout was further supported by Dimitriu et al. who also assessed burnout using the Maslach Burnout Inventory- Medical Personnel (MBI-HSS(MP)) [12]. One hundred medical residents were sent questionnaires including the 22 questions from the MBI. On average, 76% of the sample reported burnout. The authors noted that this level of burnout was “superior to studies conducted in normal periods” [12].

When Wu et al. further questioned them about their main worries and stress, 64% reported that they worried about becoming infected and 76% were worried about their families contracting the virus [7]. Both Wu et al. and Dimitriu et al. also found that there was a higher prevalence of burnout syndrome in staff working in regular wards compared to those working on the front line [7,12]. This idea will be further discussed in 5.1 Occupational Factors.

Guisti et al. also found that several personal and work-related factors contributed to the level of burnout [13]. The main reported predictors of burnout amidst this pandemic included:occupational factors such as department of work,female gender,increased work hours.

The above-mentioned predictors of burnout will be further evaluated in Section 5—potential factors contributing to physician burnout amidst the COVID-19 pandemic.

### 3.2. Other Survey Instruments to Measure Physician Burnout during COVID-19

Two other studies have reported the impact of COVID-19 on physician burnout, however, they have used other survey instruments other than the Maslach Burnout Inventory.

The first study by Morgantini et al. analysed 2707 responses from healthcare workers across 66 countries on their experiences [15]. Assessment of burnout was indicated by a “single item measure of emotional exhaustion” [15]. Hence, depersonalisation and personal accomplishment were not taken into account. More than half, 51.4%, of respondents reported burnout purely due to their work circumstances. The results found that adequate personal protective equipment (PPE) was a protective factor against reported burnout. The lack of PPE has been raised in many different countries and has led to significant levels of frustration [15]. Contrary to the findings from Wu et al. and Dimitriu et al., Morgantini et al. found that burnout was associated with exposure to COVID-19 patients (95% CI = 1.05–1.32, *p* = 0.005) [15].

The second study by Kannampallil et al. included a web-based survey which was sent to 1375 US physician trainees that assessed burnout using the Stanford Professional Fulfillment Index (PFI) [14]. The three components of burnout measured using the MBI, emotional exhaustion, depersonalization and personal accomplishment, correlates with the PFI, however, the authors’ decision to use PFI over MBI was due to the fact that the questions from the PFI captured burnout in the “past two weeks” [14]. There was a 29% response rate, and the sample was broadly split into “exposed group” and “unexposed group” accordingly, based on their response to the question about caring for patients currently being test for COVID-19 [14]. The results overall found certain predictors for burnout, which further supports the earlier mentioned predictors of burnout: female gender, increased work hours and family concerns. Doctors have often expressed the risks to their families but also one can recognise the feeling of being ‘alone’, with no one to turn to for help. Existential questions about what will happen to their families if they fall ill are bound to be prominent fears. There are a few studies that have addressed this question, but this will likely come up in the next few months.

Emerging papers [22] further reinforced these finding from listening sessions that were conducted with healthcare professionals. Along with inadequate access to PPE, other causes of anxiety leading to burnout were thought to be lack of access to up to date information and communications and unknowingly bringing COVID-19 infection home.

## 4. Potential Factors Contributing to Physician Burnout Amidst the COVID-19 Pandemic

There are a variety of factors potentially contributing to burnout during the COVID-19 pandemic. However, due to our current situation of the ongoing nature of this pandemic, there is still a lack of evidence of proven factors contributing to burnout during COVID-19. Hence, we can use proven factors contributing to burnout prior to the pandemic and our existing knowledge on physician burnout amidst this pandemic to help guide us best in providing support for physicians and healthcare professionals working during these unprecedented times.

### 4.1. Occupational Factors

The first concept which was highlighted was the department in which an individual works. Pre COVID-19, it was highlighted that burnout rates were highest amongst physicians involved in frontline care [6].

To determine whether the same correlation was seen during the pandemic, Wu et al., in addition to finding that overall physicians experienced more burnout, also compared the frequency of burnout between those physicians working in frontline wards and those working in usual wards in China [7]. They found that medical staff working on the front line had a lower frequency of burnout compared to those working on usual wards. Data from the study showed that the frequency of burnout in physicians working in frontline wards, 13%, was significantly lower than those working in usual wards, 39% [7]. This study then continued to explain this unexpected trend, suggesting that frontline workers may have felt a greater sense of control of the situation, as their job deals with uncertainty all the time, hence, this could be the reason why they are experiencing less burnout during this pandemic [7].

This correlation was further supported by Dimitriu et al., who found that 86% of normal ward workers had reported burnout compared to a prevalence of burnout of 66% in workers of frontline departments [12]. Dimitriu et al. further went on to explain a similar theory to that suggested by Wu et al. of a greater sense of control over the situation [12].

On the contrary, Kannampallil et al. found that the sample of physicians who were exposed to COVID-19 tested patients had a higher prevalence of burnout (46.3%) compared to those who were not exposed (33.7%) [14]. The authors went on to explain that the reasoning behind this could be that the non-exposed group were doing remote work. [14]

### 4.2. Gender Difference

When it comes to which gender is most affected by burnout, there are unfortunately very few studies. Pre COVID-19, there have been contrasting results with some studies finding no gender differences whereas other studies found that female surgeons experienced more burnout compared to male counterparts [2]. Prior to COVID-19, one such study by McMurray et al. found that women had increased odds of reporting burnout when compared to men. Furthermore, this study highlighted that lack of workplace control was a predictor for burnout in women but not in men [3]. Papers reported by Koh et al. and Maunder et al. both suggest that having children is a predisposing factor to burnout [4,5]. However, McMurray et al. found that women physicians who had young children to look after reported a decrease in burnout by 40%, if there was a spouse, supporting colleague or significant other to balance work and home issues [3]. These studies were done before the pandemic and as such one needs to be careful when interpreting its replicability.

There is little research on the experience of burnout with different genders during this current pandemic. However, Kannampallil et al. found that there was a higher prevalence of burnout amongst women and unmarried trainees [14]. The authors then went on to explain that specific stressors for burnout during this pandemic for women included childcare and work-life balance [14].

In addition to our reviewed articles, we found a group of researchers from Boston Consulting Group who sent out a survey to more than 3000 people in Europe and the US. Results from this survey suggested that after this pandemic, women are spending an extra average of 15 h on unpaid domestic labour compared to men [23]. A further study by the University of Melbourne found that working parents are having to care for an additional 6 h, with women taking “more than two-thirds of that extra time” [24].

With the ongoing nature of this pandemic, one can only speculate that the lasting impact of burnout for female healthcare workers will be considerable. One could hypothesize as mentioned above that the uncertainty of when the pandemic will end, if indeed it does, is a factor.

### 4.3. Increased Workload/Sleep Deprivation

Previously, sleep deprivation has been identified as a key risk factor for burnout in physicians [17,18]. With the ongoing pandemic, undeniably some of the current challenges that physicians are facing include high workload/long work, hours which ultimately can impact one’s sleep. One recent study, prior to the pandemic, of 959 healthcare employees found that 33% of the respondents were screened as positive for at least one sleeping disorder [17]. The prevalence of each sleeping disorder was reported accordingly: insomnia (17%), obstructive sleep apnoea (14%) and shift work sleep disorder (11%). The study authors also assessed burnout using the Maslach Burnout Inventory-Human Service Survey (MBI-HSS) and noted that screening as positive for a sleep disorder was associated with 4-fold increased odds of burnout [17]. Hence, it is imperative to evaluate the effectiveness of potential measures that may help promote healthy sleep, with the objective of reducing burnout and its negative effects, particularly during this current pandemic where physicians may be at an increased risk of sleep deprivation.

## 5. Discussion and Potential Support for Physician Burnout

The major focus during this pandemic has been on addressing the acuity of patient presentation, containment, preventing spread or at least limiting the spread of the virus. While this is certainly important from the point of view of pandemic management, the needs of healthcare workers are something that needs to be addressed. One only needs to consider that most places are in the first wave with the possibility of second and third waves, therefore, the likelihood of added stress on physicians needs to be kept in mind. Although there are very few papers to substantiate current levels of burnout, the emerging impact of COVID-19 and prevalence of burnout should raise urgency with which we should address burnout amongst physicians

Burnout may appear to be less frequent among frontline workers compared to usual ward workers [7,12], however, there is still a staggering prevalence of burnout in general amongst physicians compared to non-COVID times. A recurring theme of a sense of control amongst frontline workers in dealing with the pandemic was evident. Hence, it may be important for upcoming physicians to have early training on pandemic planning and incorporate burnout management techniques. Burnout caused by occupational factors such as the department an individual works in may be inevitable, hence, management of burnout must be considered. A study by Amanullah et al. showed that a hospital-based programme using mindfulness helped reduce the impact of organizational change on physicians. The authors saw the role of a mindfulness-based programme as being positive [20]. Physicians who took part reported that they handled burnout better. While the costs were minimal, the outcome clearly showed that we are not helpless in challenging situations. This was further supported by Krasner et al., who also found that self-awareness and mindfulness have been shown to effectively reduce burnout [21]. This study showed that while some took the time to learn how to be mindful, the results were evident to those who stayed the course. Being busy was often cited as a reason for not being able to be part of the study [21].

In addition to the department that an individual works in and the lack of support from peers, Sansongahar et al. reported other occupational hazards with exposure to COVID-19 including “limited resources, longer shifts, and disruptions to work-life balance/sleep”, which have been reported to increase physicians’ burnout levels [9]. The lack of PPE has been correlated with an increase in burnout [15], hence, Santarone et al. recommends that providing adequate PPE should be top priority [10]. In addition, these authors referenced one study that showed “limiting shifts to less than 16 h” resulted in an “18% reduction of attention failures” [25]. Hence, manageable shifts should be timetabled for physicians. It is imperative that periods of rest and relaxation are given to physicians to prevent burnout. With manageable shifts put into place, sufficient sleep takes priority since sleep deprivation has been linked with burnout [17,18,19]. Stewart et al. recommend early detection and intervention to improve both sleep deprivation and burnout [18].

Another important aspect of burnout, as reported by McMurray et al., is that when physicians feel they are supported by each other and at home, the incidence of burnout is less [3]. In this study, they found that support by a spouse decreased burnout by 40% and support from colleagues decreased burnout by 45% [3]. Shanafelt et al. agree that having a partner or being married was associated with a decreased risk of burnout [6]. It is clear that physicians who are supported or feel supported by their peers or loved ones experience less burnout when compared to those who do not. We can infer that colleagues’ ability to offer help in a stressful work environment helps to reduce the burden more than just the support at home. It is clear that we need more studies to prove such a hypothesis and findings. Adapting programmes may be the way forward; this however will require the creating of hospital-based committees or physician organizations working to address acute, subacute and longer-term needs post COVID-19.

This review also found that, overall, female physicians reported increased burnout in comparison with their male counterparts [14]. As women account for a huge proportion of the healthcare workforce worldwide [26], one could speculate the impact that this pandemic has had on the mental health/burnout of working female healthcare workers to be considerable. The loss of earnings is one aspect. Paying off debt, the uncertainty of single parents about their ability to provide, added to emotional stressors if they are going through a separation, divorce, substance abuse only adds to the enormous stress being faced by female physicians. Hence, targeted support for the mental wellbeing of female physicians is a must, although there is little research-based evidence on successful support methods. However, from our review, it became evident that women may have felt a greater sense of burnout due to lack of control in their workplace [3]. In fact, from a review of data, it is apparent that systems have a duty to recognize that there should be autonomy for physicians in practice [16].

## 6. Authors Recommendations

Recommendations from the authors include but are not limited to:A formalized burnout reduction programme offered within the institution or by the organization in a safe and comfortable external space.Online or telephone access to helplines operated in rotation by trained mental health professionals. This should ideally be psychiatrists, but the shortage of psychiatrists will be a limiting factor. Hence, the ability to look at developing programmes for counsellors to recognize the unique challenges that physicians have will be helpfulSupport programmes for spouses and dependents.Training during medical school and residency education should incorporate stress management techniques and pandemic planning.Gender-based issues may need to be addressed as well as there are differences in how burnout affects men and women. Although beyond the scope of this paper, the authors came across papers that highlight that differences do exist.

It is in the best interest of public health that governments start to recognize, prioritize and address rolling out effective interventions to prevent and manage physician burnout.

## 7. Conclusions

In conclusion, burnout amongst physicians is an important issue because it not only has an impact on the physician’s life, but it can potentially affect patient care, let alone, their families and society. The current pandemic has brought with it ways of working that physicians need to adapt to, and developing ways to cope with burnout is important. The ability of hospitals to help with burnout management may be helpful and certainly more studies on burnout levels and looking at comparing data between regions and nations is needed. It may be important to learn best practices from other places and replicate it.

Future research is needed on the larger spectrum of burnout that has not been addressed in this review, which are important issues and includes but is not limited to personality, social situation and financial status. These can have a bearing on how one perceives burnout and interventions that may be sought.

## Figures and Tables

**Figure 1 healthcare-08-00421-f001:**
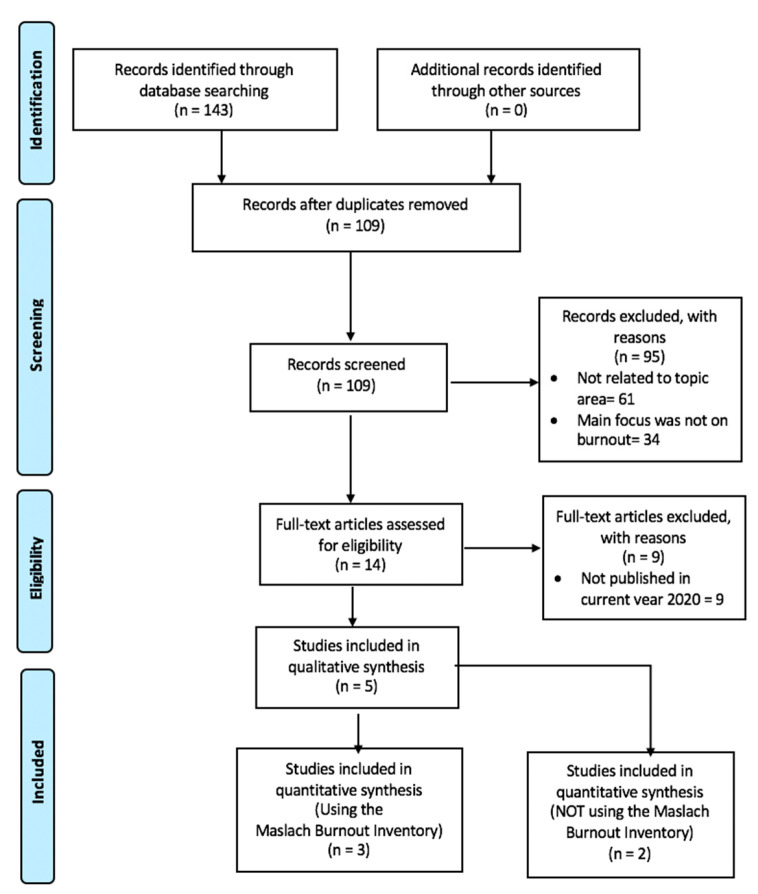
Flow diagram of selection process.

**Table 1 healthcare-08-00421-t001:** The results of the individual search terms after running through Medline and Embase (OVID) for burnout in physicians during COVID-19.

Number	Terms	Results
1	“COVID-19” OR “pandemic”	195541
2	“burnout” OR “healthcare professional burnout” OR “mental health outcomes” OR “provider burnout “OR “psychological impact”	84665
3	“healthcare professional” OR “health professional” OR “healthcare worker” OR “provider” OR “physician” OR “medical residents”	419277
4	1 AND 2 AND 3	143

**Table 2 healthcare-08-00421-t002:** Methodological characteristics and the main results for studies on the impact of COVID-19 on physician burnout.

	Author	Study Design/Sample	*N*	Results
**Using MBI**	Dimitriu et al., 2020 [11]	A cross-sectional study that compared burnout between normal ward workers and frontline workers in COVID wards using the MBI in Romania.	100	On average, 76% of the sample reported burnout. Burnout was more prevalent in normal ward workers (86%) compared to medical residents working in frontline departments (66%) (*p* < 0.05).
	Giusti et al., 2020 [12]	A cross-sectional study of health professionals working in an institution in Northern Italy were sent online surveys investigating burnout using MBI.	330	Of the sample, 76% reported burnout. Burnout was related to long work hours, fear of infection and perceived support by friends.
	Wu et al., 2020 [7]	A survey composed of the MBI was administered to physicians and nurses on the frontline (FL) wards compared with those working in usual wards (UWs).	220	Burnout was less prevalent amongst frontline staff (13%) compared to those working on usual wards (39%) (*p* < 0.0001).
**NOT using MBI**	Kannampallil et al., 2020 [13]	A cross-sectional study assessing burnout using the Stanford Professional Fulfillment Index amongst physicians training in the US.	1375	Higher prevalence of burnout in the group exposed to COVID-19 (46.3% vs 33.7%) (*p* = 0.002). Female staff and those who had longer work hours were likely to experience more burnout (*p* = 0.043).
	Morgantini et al., 2020 [14]	A cross-sectional survey of healthcare professionals (HCPs) on the front lines against COVID-19.	2707	Burnout was associated with exposure to COVID-19 patients (95% CI =1.05–1.32, *p* = 0.005).

**Table 3 healthcare-08-00421-t003:** Methodological characteristics and the main results for studies prior to COVID-19 on potential factors contributing to burnout.

Factors Contributing to Burnout	Author	Study Design/Sample	*N*	More Detail on Factors Contributing to Burnout
Occupational factors	Dunn et al., 2007 [15]	Questionnaires were sent to physicians over a 5-year period.	22–32	Emotional and work-related exhaustion decreased significantly over the period.
	Shanafelt et al., 2012 [6]	A national study of physicians from compared with a probability-based sample of the general US population.	7288	Burnout is more common among physicians than among other US workers.
Gender difference	Dyrbye et al., 2011 [2]	Cross-sectional survey of Members of the American College of Surgeons.	7858	Women surgeons were more likely to experience burnout.
	Koh et al., 2005 [4]	Self-administered questionnaire sent to Healthcare workers (HCWs) who were at the front line during the battle against SARS in Singapore.	10,511	During epidemics, more than half reported increased work stress.
	Maunder et al., 2004 [5]	A self-report survey sent to healthcare workers at three Toronto hospitals in May‒June 2003.	1557	Burnout was greater in those who cared for SARS patients. Other personal lifestyle factors contributed to burnout.
	McMurray et al., 2000 [3]	Nationally representative random stratified sample of physicians in primary and specialty nonsurgical care.	2326	Gender differences exist in burnout.
Increased workload/sleep deprivation	Quan et al., 2019 [16]	Sleep disorder screening for healthcare employees that have participated in the “SHAW program”.	959	Screening positive for sleep disorder increased risk of burnout.
	Stewart et al.2019 [17]	A review	-	Highlights emerging concepts about the role of sleep in physician burnout.
	Velo-Bueno et al. 2008 [18]	A representative sample of physicians from 70 medical centres in Spain.	240	Clear relationship between sleep deprivation and burnout.

**Table 4 healthcare-08-00421-t004:** Methodological characteristics and the main results for reviews on support/recommendations for physician burnout before and after COVID-19.

	Author	Study Design/Sample	*N*	Brief Overview
**Pre-COVID-19**	Amanullah et al., 2017 [19]	Maslach Burnout Inventory- General Survey version (MBI-GS), which is a different version of the standard Maslach Burnout Inventory, was handed to physicians of a Canadian general hospital.	55	Self-awareness and mindfulness were shown to effectively reduce burnout.
	Krasner et al., 2009 [20]	Before-and-after study of primary care physicians attending Continuing Medical Education (CME) course in 2007–2008 in Rochester, New York. The course included information on “mindfulness meditation, self-awareness exercises and discussions”	70	Participation in a mindful communication program was associated with improvements in well-being.
**During COVID-19**	Santarone et al., 2020 [10]	A review	-	Support available for burnout.
	Sasangohar et al., 2020 [9]	A review about the experience of occupational fatigue and burnout.	-	Recommendations to prevent burnout and mitigate occupational stress.
	Shanafelt et al., 2020 [21]	A review	-	Multiple factors contribute to burnout amidst the pandemic.

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
