# Peer review of "The Impact of COVID-19 on Physician Burnout Globally: A Review"

_healthcare, 2020, doi:10.3390/healthcare8040421_

Round 1

Reviewer 1 Report

Overall the changes that have been made significantly enhanced the quality of this paper. The format of the changes are also appreciated, as it made seeking out the revisions much easier.

In terms of things that need to be changed, there are some minor changes I would recommend prior to publication. The sections 5 and 6 contain excellent additional information, but there are some grammar/syntax errors that should e corrected.

There are still a few organizational issues that need to be addressed. In section 4 Guisti's research is discussed, where factors related to physician burnout are listed, but then that information is not discussed in section 5, which is called "Potential factors contributing to physician burnout. . .". In addition, there are many things mentioned in section 5 that are repeated in section 6, but then there are things in section 6 mentioned that are not in 5. For example, PPE is mentioned in section 6 (strategies to reduce burnout), but not in section 5.  It would seem that if PPE availability contributes to the perception of burnout, that info should be in section 5.  Also, Section 5.2 you discuss that there is mixed evidence for gender differences in regards to burnout, but on line 330 you report that women have a higher rate of burnout than men.

I really like that you have changed section 7 to authors' recommendations. Perhaps it is beyond the purview of this paper, but it would be nice to have a rationale or support for each of the suggestions. So for example, instead of just saying that you should add stress and burnout training during medical school, you could have that statement and follow up with something like, "Previous research has shown that stress reduction programs during medical school have been shown to be effective but are not widely implemented (Daya & Hearn, 2018)"  Daya, Z., & Hearn, J. H. (2018). Mindfulness interventions in medical education: A systematic review of their impact on medical student stress, depression, fatigue and burnout. Medical teacher40(2), 146-153.

Overall I fell this is much improved, and with a few grammar and organizational changes, this would be a fantastic resource for those studying burnout.

Reviewer 2 Report

I realize that a great work and time has been devoted to this paper. This is a topic of great significance to emotional wellbeing of physician that can affect the quality of care provided to their patients, and therefore, be a public health problem. So I appreciate authors examining this topic and burnout syndrome.

The paper has a lot of strengths but I think that some changes should be recommended.

Abstract:

Readers should be able to read the abstract in isolation and understand what you have done, and its implications. So I recommend to the authors to expand the “Results” with more information, because it is unclear which correlations are significant. Please, include the subheadings of background, methods, results and conclusions as a structured abstract.

It is not necessary to cite the authors and references in the abstract. Please, avoid it.

Please, use “physicians” instead of “doctors”, because it refers to PhD, not in medical aspect.

Introduction:

The introduction is very scarce, is disorganized, severe lacking of references and does not justify why it has conducted this study.

The aim of this study would be a viable and feasible objective of being answered in a systematic review.

Methodology:

Please, the authors would specify the data of review, if they have followed the PRISMA recommendations, etc.

PubMed is not a database, so it’s the search engine for the database Medline.

What kind of design studies have the authors chose as inclusion criteria?

When the authors say that they have included studies that used the MBI or similar, its make no sense. The Maslach Burnout Inventory (MBI) is the most reliable test for burnout, despite of professional category and there are more than 100 instrument for measure the burnout syndrome. So please, justify these inclusion criteria and if finally, they only choose the MBI or others.

The Flow Chart does not follow the PRISMA recommendation.

Results, Discussion and Conclusion:

The table does not follow APA style.

The authors leave the full weight of the results in the tables.

The subheadings 4 and 5 are part of results or the discussion?

The results are totally devoid of data and seem more like a discussion than results. Prevalence data are lacking, explain correlations, p-value, results in terms of study variables, etc.

Conclusions should be taken with caution and drawn from articles published after the pandemic and not before.

References:

Bibliographical references do not conform to the rules of the journal.

Please, read the author guidelines.

I hope that these recommendations do not discourage the authors and I want to recommend the authors to continue working on this paper.

Reviewer 3 Report

find it a very interesting article, I think it contributes a lot on a scientific and social level. However, it has certain shortcomings in several sections of the manuscript.
Summary
The abstract should be more structured. Introduction. Objective. It is not clearly indicated which methodology has been followed, or the database, or the inclusion criteria. The results and finally conclusions must be presented.
A few studies, however, have made recommendations that may help prevent burnout and mitigate the consequences of occupational stress during COVID-19 [1,2] (Sasangohar et al. 2020, Santarone et 17 al. 2020). This is justification for the study that should appear in the introduction. It is not advisable to include authors in the abstract.
Introduction
It is necessary to go deeper (concepts, variables, previous studies, etc ... are missing) and justify why the study is going to be carried out. It is very brief.
Methodology
The inclusion and exclusion criteria are not well defined.
In what time was it done?
Has the PRISMA guide been followed? For example.
Are there risk of bias?

Round 2

Reviewer 2 Report

Dear Authors,

Congratulations on the work done and the changes made.

I think that the paper has improved.

I wish you luck!

Author Response

Dear Sir/Madam,

Thank You for reviewing our manuscript once again and for all your previous comments. We believe that our manuscript has improved significantly with your suggestions

Most sincerely,

Shabbir Amanullah (Corresponding Author)

Rashmi Ramesh Shankar (Second Author)

Reviewer 3 Report

Abstract: In method and results the same studies are repeated. The method should include the inclusion criteria of the studies.
Introduction: The introduction should have more content. Indicate which instrument is used to measure burnout, the repercussions it has on health professionals.
Be careful with the numbering in the text. There are statements that have no reference.
There are studies that determine the level of burnout among health professionals in the Covid-19 crisis.
This quote is recommended:
Ruiz ‐ Fernández, M. D., Ramos ‐ Pichardo, J. D., Ibáñez ‐ Masero, O., Cabrera ‐ Troya, J., Carmona ‐ Rega, M. I., & Ortega ‐ Galán, Á. M. (2020). Compassion fatigue, burnout, compassion satisfaction, and perceived stress in healthcare professionals during the COVID-19 health crisis in Spain. Journal of clinical nursing.
Results: Very difficult to follow up the results, I advise to sort a little headings, tables, figures. Perform numbering with table 1, table 2,… .etc.

Author Response

Dear Sir/Madam,

Most sincerely,

Shabbir Amanullah (Corresponding Author)

Rashmi Ramesh Shankar (Second Author)

This manuscript is a resubmission of an earlier submission. The following is a list of the peer review reports and author responses from that submission.

Round 1

Reviewer 1 Report

The article, “The Impact of COVID-19 on Physician Burnout Globally: A review” is a very timely paper to address the possibility that COVID will increase physician burnout. It provides a comprehensive listing of articles related to physician burnout, and it also provides several recommendations on how to address the enormous problem of physician burnout.  While this is a good article that I will definitely use as a resource, there are some items (both general and specific) that should be addressed.

General: This article has many places when things are not stated clearly, and sentences have poor phrasing.  There are also some conceptual problems that need to be resolved. This includes clarifying the concepts being discussed and not using burnout and stress and anxiety interchangeably, as if they are the same terms.  In addition, any time concepts are mentioned there should be support offered for those concepts. It seems as if there is much about COVID that is taken on assumption but is not supported.  Does it result in more stress? More work hours?  Decreased autonomy?  Decreased sleep?  These things are all assumed but not supported.  Another aspect that should be addressed is that burnout has three main components (depersonalization/detachment, emotional exhaustion, and ineffectiveness/lack of accomplishment). These components are only mentioned once (in the gender discussion), and each of these could play different roles in burnout related to COVID.  This is a giant gap in the current paper.

The major tenet of this paper is that burnout is a problem in physicians and that COVID is going to make this worse. But Wu’s article and the results you report (paragraph starting on line 98) seem to suggest that burnout got better during COVID for both front line and usual workers.  You don’t ever address this again or use the information about WHY this might be so in your recommendations.

Specific Comments:

  • In the first paragraph you describe how the current COVID pandemic has made things more difficult for medical professionals, but you don’t say why (lines 31 and 32). While it is fairly obvious why this may be true, you should include some information about how physician lifestyles, workload, stress, burden, and interactions with patients have changed due to COVID.
  • Be consistent on how you write Physician Burnout (34) or Physician burnout (41)
  • Line 47: Google Scholar and PsychLit are proper nouns (cap)
  • Line 46 OVID, line 51 Ovid (be consistent)
  • Line 54 citation for MBI reliability? And did you mean to say valid instead of reliable?
  • Table 1: I am confused why some studies have missing information (Adams 2020, etc).
  • Line 64: This statement is confusing. A number of studies about what?
  • Line 98: add year to citation
  • Line 99 Front line; line 102 frontline (be consistent)
  • Line 121: anxiety/burnout levels are mentions, but these are two very different concepts (and there is no citation). If you want to address both, please separate them and then offer supporting evidence for both.
  • Line 132: two periods at end of sentence
  • Line 143: do you have any evidence that there are changes in sleep habits/patterns/quantity/quality during the pandemic?
  • Line 166 throws in age effects of the pandemic, but it is in the gender paragraph, and it seems incomplete. I would either leave this out, or ideally have a section dedicated to age differences
  • Line 179: year of citation?
  • Paragraph beginning at line 179: this paragraph starts with a sentence about how addressing burnout is achievable, but the rest of the paragraph addresses reducing stress in response to organizational change with no further reference to burnout.
  • Line 185: I think the word incidence is better than frequency in this situation
  • Line 188: May be (not maybe)
  • Line 193 and Line 194 seem more like opinion and less supported by research.
  • Recommendations: paragraphs 2, 3 and 4 in this section seem a bit disorganized. You discuss the benefits of support, but you mention spousal support in paragraph 2, then mention it again with more evidence in paragraph 3 without linking them and organizing. In the 3rd paragraph you suggest that isolation due to COVID will affect support, but you offer no evidence that there are changes in spousal or co-worker support during COVID.  This section seems a bit weaker than the rest.
  • Paragraph starting line 212: you start this paragraph talking about autonomy but then seem to get distracted talking about burnout at the health systems level.  More support for what autonomy is and why it matters would be beneficial.
  • Line 224: this is the first time you mention educational materials, interventions and call centres.  Should this be included earlier to build support for it in the conclusions?
  • Line 225: include but ARE not limited to
  • The discussion and conclusion section confuses me, as it brings up a listing of recommendations that are not mentioned or supported in their article. I am confused about whether these suggestions are based on the articles in the review or if they are from the authors, and I also want to see support for these. Where did these come from, and why?
  • The recommendations/discussions/conclusions do not really break down anything the way your article did. Do you have specific recommendations for front line vs usual workers? Females vs males?

Reviewer 2 Report

Introduction

re: the Shanafelt reference.  this sounds like a lifetime prevalence of burnout?  Please clarify.   

“physician burnout” shouldn’t be capitalized

Methods

You don’t need to call it a “computer-based” search, we can assume that. 

You report that you “tried” to include articles that assessed burnout using the Maslach Burnout Inventory.  Did you actually restrict the articles to only those which used this scale?  You planned to compare the results of these papers?  why have you included reviews here.  

Instead of showing the MeSH search you should start with the 180 articles and start a flow sheet of the articles which were filtered based on the exclusion criteria you set.

It may help for you to describe what you plan to do with the articles you have chosen.  perhaps that you will review burnout in general and then apply this understanding to the more recent articles that describe burnout in the time of COVID?

Results

The table belongs in the results.  Perhaps organize with landscape orientation given all the text.  I would re-organize the subsections inside the results and include all the articles from your review.

1. Factors contributing… I would start with this section and focus on articles prior  to COVID.  

2. Physician Burnout during COVID-19.  “There have been a number of studies and one study”.  It appears that you are reviewing two articles here?   I would comment on all the articles using the MBI scale in the time of covid.  Perhaps allow the prior sections which reviewed past research on burnout to inform this section.  

3. I agree with ending with the recommendations

Is it true that not a single article in your review commented on vacation or a break from work as a way to combat burnout??  The definition of burnout is psychological exhaustion related to work.  If vacation or a break from work was not studied in the context of pandemics or COVID-19 I would add a comment that in all these articles this very simple antidote for burnout (which has been described in other literature) was not examined for unclear reasons. 

Reviewer 3 Report

The subject of the paper under review, physician burnout during the current pandemic, is an important issue that is starting to draw the attention of researchers and decision makers. However, the paper under review does not fulfil the established criteria to be accepted for publication, It is written without due care and has several flaws. These can be grouped under four main headings: (a) the search methodology for the papers under review is not rigorous and clear, as evidenced by the heterogeneity of the 21 studies the authors present, (b) the text does not follow a strict and structured process of presenting the data from the papers reviewed, separately from the discussion of the results, (c) the recommendations in the “Discussion and conclusion” section are not connected to the paper’s findings, but seem to be a list that reflects the authors’ opinions and bias and (d) the references do not conform to the journal guidelines.

More specifically:

  • Search methodology:
    • The authors present a three step approach to select the papers for inclusion in the review, the first of which is "COVID-19" OR "pandemic" OR "coronavirus". Yet some of the 21 papers, published before 2020, do not include the terms “pandemic” or “coronavirus”, e.g. Amanullah et al, 2017, Linzer et al, 2000, Shanafelt et al, 2012.
    • It is stated that “In the end, only 21 studies met full criteria and were included in this review” – but it is not stated clearly what other criteria were applied so that from 180 studies only 21 are included.
    • It is stated that “We tried to include articles that assessed burnout using the Maslach Burnout Inventory to make the results comparable”, yet only very few of the 21 papers use the mentioned questionnaire!
    • The usual purpose of a step-wise selection of papers for inclusion in a review is that the selected papers have similar methodology so that the results can be analysed and synthesized. In this case, some papers are research papers, where data was collected using a questionnaire, and others are reviews, while one (Santarone et al) is more of an opinion article.
  • The contents of the paper
    • In a systematic review process, the analysis and presentation of the papers that are selected should be clearly separated from the discussion and synthesis of the results. In the paper under review, there is usually a paragraph at the end of several subsections, with a conclusion that the authors draw from the papers presented. Examples: p5 line 87 “It is indisputable…”; p5 line 109, “the major focus…”; p7 line176 “Lack of supports…”  These conclusions should be separated and presented together in the discussion section.
    • Also, the data from the 21 papers selected should be separated from findings from other sources. In the paper under review, the findings of other papers, publications and internet sources are mixed with the findings of the 21 papers that were selected, e.g. the references to BCG (Boston Consulting Group), to the Council on Foreign Relations, to the BBC (British Broadcasting Corporation). Furthermore, some of the additional sources are not peer reviewed papers, and therefore cannot be presented as being of equal value with the peer reviewed papers.
    • The section entitled “Physician Burnout during COVID-19” and the subsection entitled “Front line workers vs Usual work workers” essentially present the findings form the paper by Wu et al twice over.
    • The subsection on “limited organisational support” has one sentence on this topic and the rest is concerned with sleep deprivation – which would thus warrant a subsection on its own.
    • In the section entitled recommendations there is a presentation of two programmes for support (one by the first author of this paper) and then goes on to discuss PPE and other issues. In this section the sentence “That the isolation post COVID will certainly exact a toll, will likely emerge in upcoming studies” is an opinion of the authors that is not supported by any evidence in the paper.
  • The “Discussion and conclusion”
    • The section entitled “Discussion and conclusion” has a list of 7 recommendations that reflect the authors’ opinion and biases, without being supported by the preceding presentation and analysis.
  • The references do not follow the journal’s guidelines.
    • In the paper, references are quoted by [Author, date], not by numbers [1,2,..] in square brackets. As a result, the references at the end are listed in alphabetical order, not in a numbered sequence.
    • The references to sources from the internet are listed by the title of the web page – even when such pages have an author, e.g. “Easing the COVID19 burden…” is by Matt Krentz.

The following reference is an example of unacceptable, careless writing: "Stress And Burnout Warning Over COVID-19". 2020. The British Medical Association Is The Trade Union And Professional Body For Doctors In The UK.. https://www.bma.org.uk/news-and-opinion/stress-and-burnout-warning-over-covid-19.”